# Three-Dimensional Visualization System with Spatial Information for Navigation of Tele-Operated Robots

**DOI:** 10.3390/s19030746

**Published:** 2019-02-12

**Authors:** Seung-Hun Kim, Chansung Jung, Jaeheung Park

**Affiliations:** 1Graduate School of Convergence Science and Technology, Seoul National University, Seoul KR 08826, Korea; ksh1018@snu.ac.kr; 2Intelligent Robotics Research Center, Korea Electronics Technology Institute, Bucheon KR 14502, Korea; chansung0602@keti.re.kr; 3Advanced Institutes of Convergence Technology, Suwon KR16229, Korea

**Keywords:** 3D visualization, wrap around view monitoring, robot vision systems, tele-operated robots

## Abstract

This study describes a three-dimensional visualization system with spatial information for the effective control of a tele-operated robot. The environmental visualization system for operating the robot is very important. The tele-operated robot performs tasks in a disaster area that is not accessible to humans. The visualization system should perform in real-time to cope with rapidly changing situations. The visualization system should also provide accurate and high-level information so that the tele-operator can make the right decisions. The proposed system consists of four fisheye cameras and a 360° laser scanner. When the robot moves to the unknown space, a spatial model is created using the spatial information data of the laser scanner, and a single-stitched image is created using four images from cameras and mapped in real-time. The visualized image contains the surrounding spatial information; hence, the tele-operator can not only grasp the surrounding space easily, but also knows the relative position of the robot in space. In addition, it provides various angles of view without moving the robot or sensor, thereby coping with various situations. The experimental results show that the proposed method has a more natural appearance than the conventional methods.

## 1. Introduction

Tele-operated robots that can work on behalf of humans have been actively studied since the Fukushima nuclear disaster. The Defense Advanced Research Projects Agency Robotics Challenge (DRC) has also been held to evaluate the performance of these robots. However, it is still difficult to remotely control robots because many factors must be considered (e.g., narrow spaces, obstacles, road or terrain conditions, weather effects, communications, visible data about the surrounding environment). One of the important factors is visible data. The surrounding environment of the robot must be constructed, and a variety of viewpoints must be visualized in real-time.

One of the methods used to visualize the robot environment is simultaneous localization and mapping (SLAM). SLAM is an algorithm used to estimate the map and the location of a robot using the surrounding information obtained from the robot in an unknown environment. Various sensors are used to obtain the information about the surroundings of the robot for SLAM. Depth sensors [1] (e.g., laser scanners, infrared sensors, lidars, and radars) and vision sensors [2,3,4] (e.g., mono cameras, stereo cameras, and RGB-D cameras) are typically used for SLAM. In visualizing the environment using these sensors, the angle of view is not 360°. Hence, the person or the robot must rotate and move to accumulate the surrounding data [5,6,7]. In addition, accumulated error can happen in an area with featureless walls or floors.

Another visualization method is the around view monitor (AVM), which provides the driver or the tele-operator with a virtual bird’s-eye view using multiple fisheye cameras. The AVM is mainly applied to vision systems (e.g., parking and tele-operation) where real-time visualization is required for safety or convenience [8,9]. The AVM presents all objects on the floor under the assumption, and hence an unnatural image is generated when obstacles are present. To solve this problem, studies are being performed to improve the user visibility using depth sensors [10].

The wrap-around view monitor (WAVM) [11] has recently been studied to provide the driver or the tele-operator with various viewpoints as well as a bird’s eye view. The WAVM maps the images to a fixed three-dimensional (3D) model to visualize the surroundings of the vehicle or the robot. It provides a large field of view and various viewpoints different from the AVM, which provides only a bird’s eye view. Various viewpoints are advantageous to the tele-operator because they provide appropriate points of view according to various situations. Therefore, the WAVM is also applied to safety monitoring systems for construction equipment. Existing WAVM studies have employed 3D models, such as hemisphere (Figure 1a) [12,13,14] or cylinder [15], based on the outdoor environment. The real world differs from these 3D models, and hence the existing WAVM practically provides tele-operators with unnatural images, which can make them feel uncomfortable because it is difficult to perceive the space around the robot.

This study proposes an improved WAVM with a spatial model using four fisheye cameras and a 360° laser scanner. The spatial model is constructed using the spatial information data obtained from a 360° laser scanner. The 360° laser scanner is a widely used sensor for gathering information about the space around a robot [16]. This provides an image of sufficient visibility for operation of the robot, and allows the tele-operator to know the relative position of the robot in space (Figure 1b). It also provides the environment of the robot at various viewpoints without moving the robot or the sensor.

## 2. Proposed Method

Figure 2 shows the schematic of the proposed system consisting of three processes, namely, image stitching, spatial modeling, and image mapping steps.

### 2.1. Image Stitching

This process is similar to the existing WAVM [9,10], and consists of undistortion, region of interest (RoI) cropping and warping, and blending steps. Image undistortion as well as RoI cropping and warping steps are performed on each image obtained from four cameras. The blending step combines four images to create one image.

#### 2.1.1. Undistortion

The fisheye camera provides an image with a horizontally wide-angle view (over 180°), but it has radial and tangential distortion (Figure 3). The four images with distortion are corrected in this step [11]. Figure 4 shows the geometric model assuming that the lens shape is spherical. The coordinate of the image with distortion is (*x_d_*, *y_d_*), while *F* is the focal length and the radius of the sphere [12]. *L* is the distance between the center point of the lens and the corrected point (*x_u_*, *y_u_*) calculated by (2). Accordingly, (1) is obtained using the ratios of the triangles shown in Figure 4, and (3) and (4) for backward matching are obtained by substituting (2) into (1). Applying the backward mapping to the entire image, the undistorted images are obtained as shown in Figure 5.
(1)$$\left(x_{d},\;y_{d}\right)=\left(x_{u}F/L,\;y_{u}F/L\right)$$
(2)$$L=\sqrt{x_{u}{}^{2}+y_{u}{}^{2}+F^{2}}$$
(3)$$x_{d}=x_{u}F/\sqrt{x_{u}{}^{2}+y_{u}{}^{2}+F^{2}}$$
(4)$$y_{d}=y_{u}F/\sqrt{x_{u}{}^{2}+y_{u}{}^{2}+F^{2}}$$

#### 2.1.2. RoI Cropping and Warping

Not all regions of the undistorted image are needed for the image stitching, because distorted regions still exist. The two RoI images with a valid meaning in the whole image are cropped out in the cropping step. The valid images are the regions in Figure 5a,b, defined as RoI A and B images, respectively. The RoI A image is mapped to both the floor and the wall because the region of the wall occupying the image differs depending on the distance between the wall and the camera. The RoI B image is always mapped to the floor.

The cropped images are warped using a homography matrix. Homography is the most common model used to describe the two-dimensional (2D) image transformation relationship of planar objects. Homography is defined in a homogeneous coordinate system. Equation (5) is a matrix of perspective transformations on a homogeneous coordinate system.
(5)$$w\left[\begin{array}{c} x^{\prime }\\ y^{\prime }\\ 1 \end{array}\right]=\left[\begin{array}{ccc} h_{11} & h_{12} & h_{13}\\ h_{21} & h_{22} & h_{23}\\ h_{31} & h_{32} & h_{33} \end{array}\right]\left[\begin{array}{c} x\\ y\\ 1 \end{array}\right]$$

The scale of homography cannot be determined because (*x*, *y*, 1) and (*wx′ wy′*, *w*) are homogeneous coordinates. Therefore, *h*_33_ is set to 1, and the remaining eight unknowns are calculated. We use a checkerboard and four matching pairs to calculate the homography matrix. Figure 6 shows the cropped and warped result images of Figure 5a,b. We determine the relationship between the source image and the result image in this step.

#### 2.1.3. Blending

In this step, two blended and stitched images are obtained using eight cropped and warped RoI A and B images in Figure 6. At this time, the region where the two images overlap appears unnatural. We blend this region to make it look natural by assigning different weights to each pixel according to the distance in the overlapping regions [13,14]. The value of pixels is computed with (6) [15], which loops over each row in the warped image:(6)$$I_{w}\left(x_{i},y_{i}\right)=\left(d_{i}/l\right)\cdot I_{1}\left(x_{i},y_{i}\right)+\left(1-d_{i}/l\right)\cdot I_{2}\left(x_{i},y_{i}\right)$$ where *l* is the length of the line that overlaps each row, and *d_i_* is the distance from the starting overlapping pixel to the current pixel *x_i_*. *I*_1_ and *I*_2_ denote the two images to perform image blending. *I_w_* is the weighted sum. The longer *d_i_* is, the greater the weight. Figure 7 shows the stitching results for RoI A and B images.

#### 2.1.4. Acceleration for Image Stitching

The above-mentioned steps require multiple matrix operations on the number of pixels in all source images to obtain two images (Figure 7). This is a factor that increases the computational time. To overcome this, we employ a lookup table (LUT) [16] instead of matrix operations. LUT denotes the data structures in computer science that are typically associative arrays, and are often used to replace runtime calculations with simple array indexing. Fetching values from memory is faster than through computation or I/O functions. In the proposed system, the relationship between the input and result images is fixed and stored in the LUT. Through the LUT, the pixel values of the result images are immediately known as the pixel values of the input images.

### 2.2. Spatial Modeling

The spatial modeling process is constructed using the spatial information data of the space. This process consists of gathering spatial information data and converting 2D points to 3D points.

#### 2.2.1. Gathering Spatial Information Data

In this step, the measurement data are first obtained using a 360° laser scanner to make the 2D spatial information. Figure 8 shows the coordinate representation according to the measurement points (*p_xn_*, *p_yn_*) calculated by the spatial information data, which are distances *d_n_* and angles *θ_n_* in (7) and (8) on the 2D coordinate. *O* indicates the position of the center of the laser scanner.
(7)$$p_{xn}=d_{n}\cdot \sin(\theta_{n})$$
(8)$$p_{yn}=d_{n}\cdot \cos(\theta_{n})$$

The measurement points may have noise caused by the diffused reflection. Hence, we use a noise filter that is robust, locally weighted, and scatterplot-smoothing [17].

#### 2.2.2. Converting 2D Points to 3D Points

In this step, two 3D points are generated at one measurement point obtained in the sampling spatial information step, assuming that the wall and the floor are vertical. The *z*-axis values of the 3D points are generated using (9) and (10) to make the 3D spatial model:(9)$$p_{zbn}=B\cdot D\cdot d_{n}/\sin(\theta_{n})$$
(10)$$p_{ztn}=T\cdot D\cdot d_{n}/\sin(\theta_{n})$$ where *p_zbn_* and *p_ztn_* are the *z*-axis values at the bottom and top points of the spatial model, respectively. *B*, *T*, and *D* indicate the height of the boundary between the wall and the floor, the height of the boundary between the wall and the ceiling, and the distance between the robot and the walls, respectively. *P_tn_* = (*p_xn_*, *p_yn_*, *p_ztn_*) and *P_bn_* = (*p_xn_*, *p_yn_*, *p_zbn_*) on the 3D coordinates are obtained by (7)–(10) as shown in Figure 9.

### 2.3. Mapping the Stitched Image to the Spatial Model

This process involves mapping the resulting image obtained by the image stitching process to the spatial model obtained by the spatial modeling process. First, the stitched RoI B image in Figure 7b is mapped to the floor region (yellow quadrangle) of the spatial model, as shown in Figure 11a. The stitched RoI B image is used without further conversion. Second, the image *I*, which is the stitched RoI A image in Figure 7a, is split into *N* images (Figure 9). Image *I* is expressed as *I* ⸧ {*I*_1_, *I*_2_, … , *I_n_*, … , *I_N_*}. *N* is the resolution of the 360° laser scanner. For example, 720 split images are obtained if *N* is 720. The split images are sequentially mapped to the walls and floors of the spatial model. When mapping the image, it is divided into a far wall and a near wall. It is a near wall if *d_n_* is lower than *D*. It is a far wall if *d_n_* is higher than *D*. In the case of a near wall, the split image *I_n_* is mapped to the wall region (blue region) of the spatial model (Figure 11b). In the case of a far wall, the split image *I_n_* is mapped to the wall region (upper blue region) and the floor region (lower red region) divided by the two points *P_Bn_* and *P_Bn_*_+1_ (Figure 10). The two points, *P_Bn_* and *P_Bn_*_+1_, are the boundary points between the wall and the floor. The image *I_n_*_+2_ in Figure 11c shows the mapping of the far wall and the floor to the spatial model. Figure 1b and Figure 12 show the result images for the spatial model for the room and the corridor, respectively.

## 3. Experimental Results

In this section, subjective and objective comparative experiments were conducted to evaluate the visibility among SLAM, the existing WAVM, and the proposed system. In our system, four fisheye cameras were mounted on the upper part of the robot such that they had the same height, as shown in Figure 13. A 360° laser scanner was mounted on the center of the four cameras. In this experiment, we used a computer with CPU 3.80 GHz and 8 GB RAM. On average, the proposed system took approximately 55 ms (18 fps) to generate the result image without a GPU for the embedded system. Processing could be accelerated if we applied parallel processing on the GPU.

### 3.1. Subjective Comparative Experiment

For the subjective evaluation, we configured a corridor experiment to compare the proposed system with the existing WAVM [10]. Figure 14 shows a c-shaped corridor drawing with a length of 40 m and a width of 3 m, as well as the location and orientation of the robot in the corridor. The yellow rectangles (**A**, **B**, and **C**) indicate the location of the robot. The arrow denotes the direction of the robot’s movement. Location **A** is an L-shaped corner in the corridor; **B** is close to the left wall; and **C** is located in the center of the I-shaped corridor. Figure 15 compares the proposed system with the existing WAVM. Figure 15a–c show the results in locations **A**, **B**, and **C**, respectively, in Figure 14. As shown in the left image of Figure 15a, it is difficult to recognize the corridor shape in the existing WAVM result image. However, as a result of the proposed system in Figure 15a (right), it is easy to recognize that the robot is in the L-shaped corridor. Likewise, the distance between the robot and the left and right walls can be easily determined, as shown in the right-side images of Figure 15b,c.

We also constructed the testbed in Figure 1b to evaluate the visibility of the existing WAVM, SLAM [17,18,19], and the proposed system. The dimension of the testbed was 6 m × 6 m × 2.1 m. Each wall had 30 markers, with a total of 120 markers. The marker measured 18 cm × 18 cm. Figure 16a shows the ground truth image taken by the camera. Figure 16 compares the result images obtained by the three methods (i.e., SLAM, existing WAVM, and the proposed system) in the same viewpoint and location. Figure 16b depicts the result image of the real-time appearance-based mapping SLAM [17,18,19] using a stereo camera. The result image in SLAM was not natural because the mismatched feature points appeared when the stereo camera moved to accumulate data. Figure 16c,d depict the result images of the existing WAVM and the proposed system, respectively. The red lines in each figure represent the lines connecting the bottom parts of the markers on the wall. The lines in (b) and (d) are straight lines, as shown in (a). In contrast, the line in (c) is a curve. The existing WAVM result image (c) was distorted because it did not know the spatial information of the testbed that the wall was vertical and flat.

Figure 17 shows the resulting images in various and complex environments. Figure 17a depicts that the surrounding environment was well represented for curved wall surfaces. In the space of 12 m or more, it was expressed as shown in Figure 17d since the visibility was better when using the bowl model than the spatial model.

### 3.2. Objective Comparative Experiment

For the objective evaluation, we used markers (blue rectangular) in the testbed (Figure 16) and adopted the root mean square error (RMSE) to evaluate the similarity to the ground truth. A total of 120 marker images were transformed with homography to calculate the RMSE. They were also cropped and normalized to the same size. The RMSE was obtained as follows:(11)$$RMSE=\sqrt{\frac{1}{h\cdot w}\sum_{x=1}^{h}\sum_{y=1}^{w}{\left(M_{g}\left(x,\;y\right)-M_{t}\left(x,\;y\right)\right)}^{2}}$$ where *M_g_* and *M_t_* indicate the marker image of the ground truth and each test image (i.e., SLAM, existing WAVM, and proposed system). *h* and *w* denote the height and the width of *M_g_* and *M_t_*, respectively.

Table 1 shows a comparison of the RMSE between the results of each method and the ground truth image when the distance between the robot and the wall was 2, 2.5, and 3 m, respectively. The RMSE values in Table 1 were the average of the RMSE with 120 marker images. As expected, the RMSE of SLAM was relatively large, which is due to feature matching errors. Furthermore, the RMSE of the proposed system was the lowest, indicating that it was the most similar to the ground truth. In summary, the experimental results of the subjective and objective evaluation showed that the proposed method had better visibility than the existing methods.

## 4. Conclusions

This study proposed a new visualization system that combines the WAVM and a spatial model to increase the visibility of robot tele-operation using four fisheye cameras and a 360° laser scanner. We compared the performance of the proposed system and the existing WAVM and SLAM that are widely used in robots by comparing experiments.

Our system provides tele-operators with a 360° view around the robot in real-time without accumulating data. This provides the tele-operator with various viewpoints and information about the changing environment, without any movement of the robot or sensor, in real-time. The tele-operator can determine the direction of the robot’s motion and prevent collision with obstacles by intuitively recognizing the surrounding space of the robot.

## Figures and Tables

**Figure 1 sensors-19-00746-f001:**
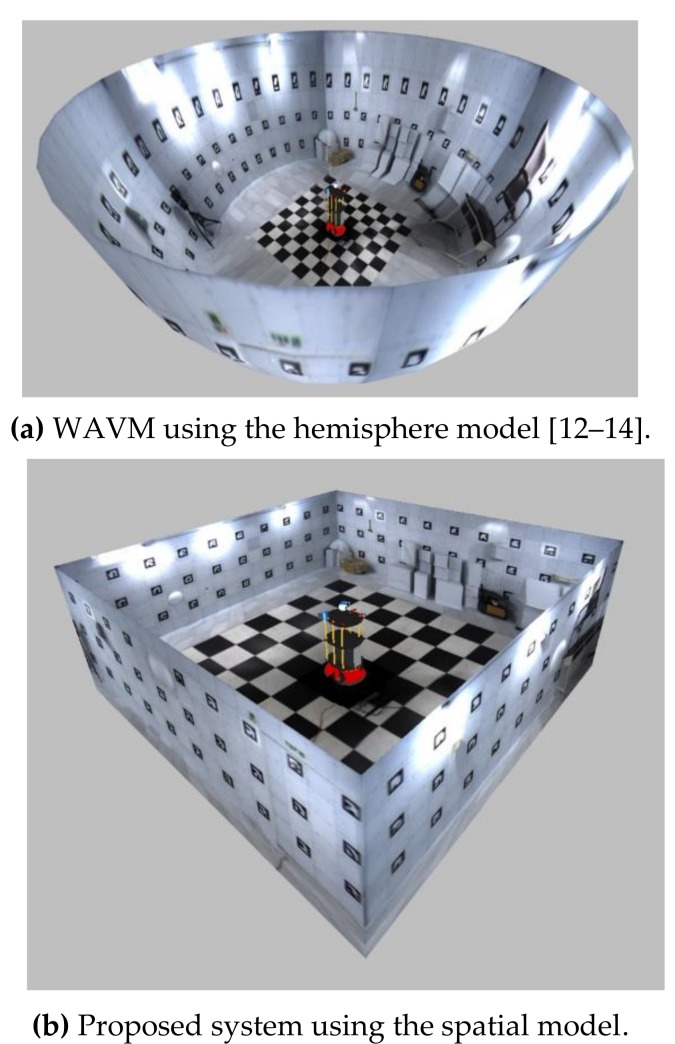
Demonstration of the wrap-around view monitor (WAVM).

**Figure 2 sensors-19-00746-f002:**
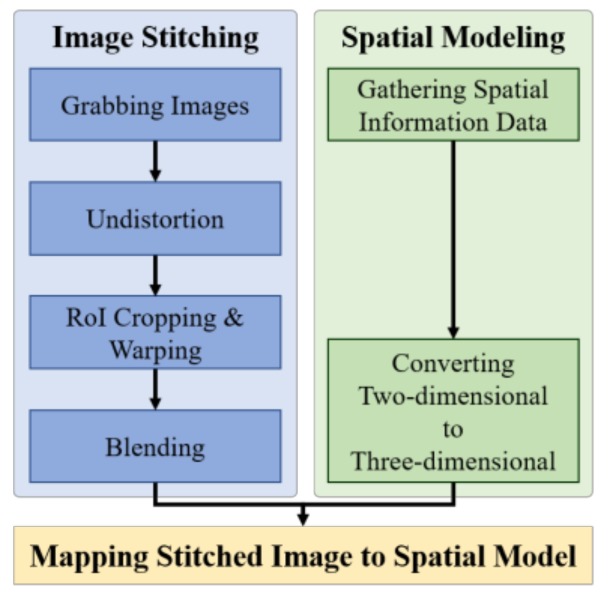
Schematic of the proposed system. RoI: region of interest.

**Figure 3 sensors-19-00746-f003:**
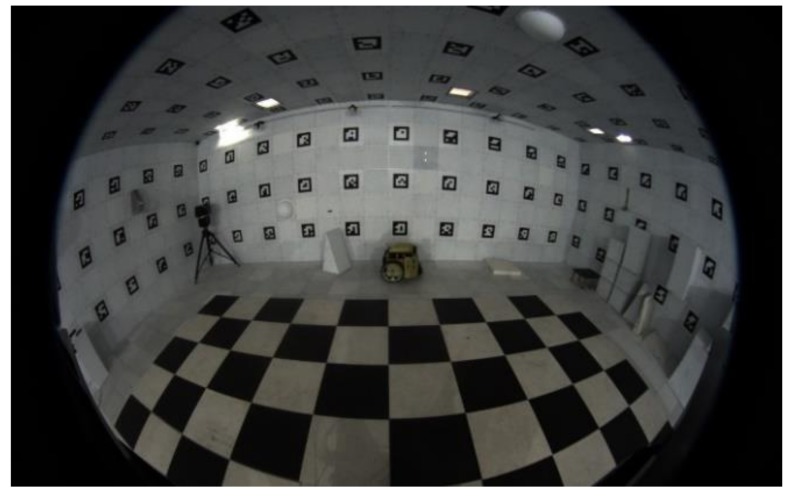
Fisheye camera image with distortion.

**Figure 4 sensors-19-00746-f004:**
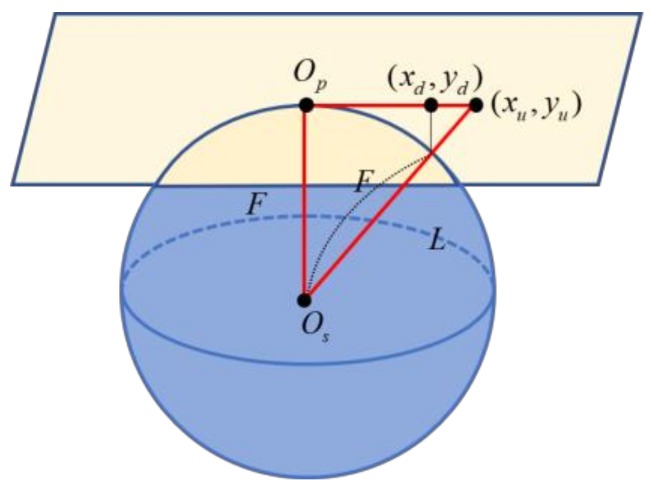
The spherical geometric model.

**Figure 5 sensors-19-00746-f005:**
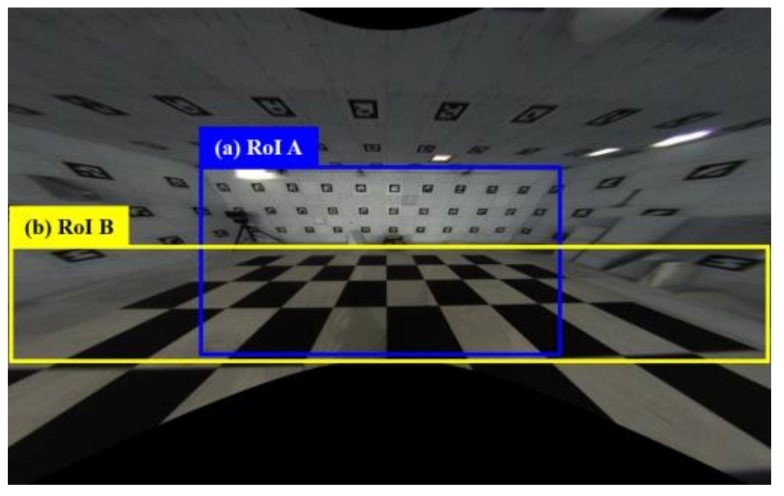
Undistorted image.

**Figure 6 sensors-19-00746-f006:**
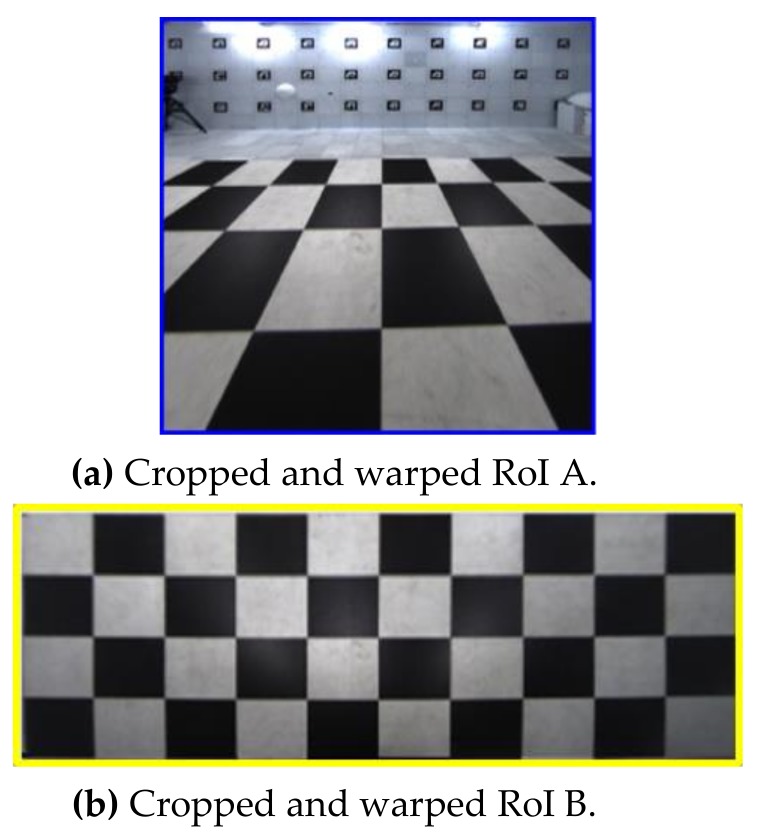
Cropped and warped images of RoIs A and B in Figure 5.

**Figure 7 sensors-19-00746-f007:**
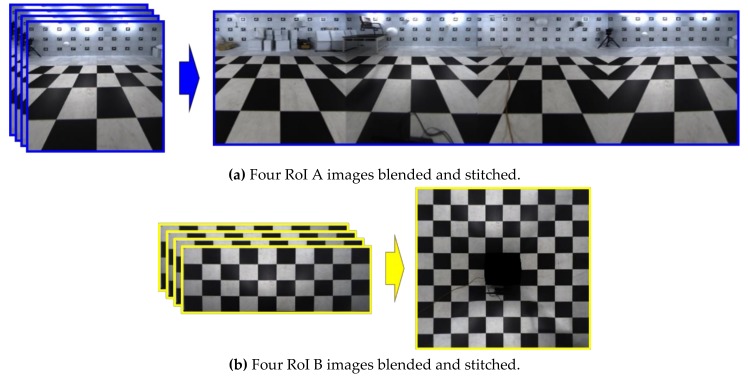
Blended and stitched RoI A and B images using the four images in Figure 6.

**Figure 8 sensors-19-00746-f008:**
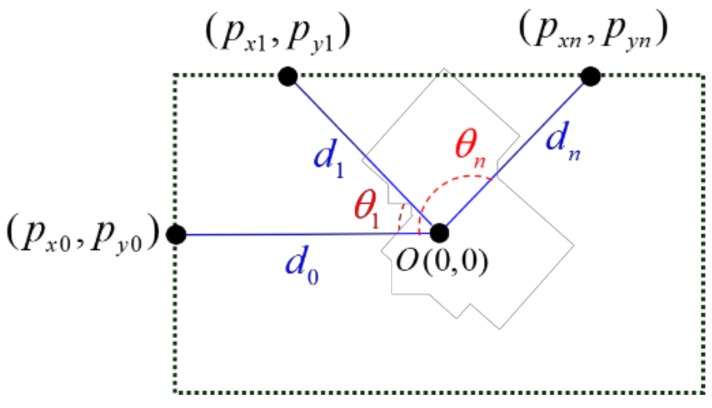
Coordinate representation according to spatial information.

**Figure 9 sensors-19-00746-f009:**
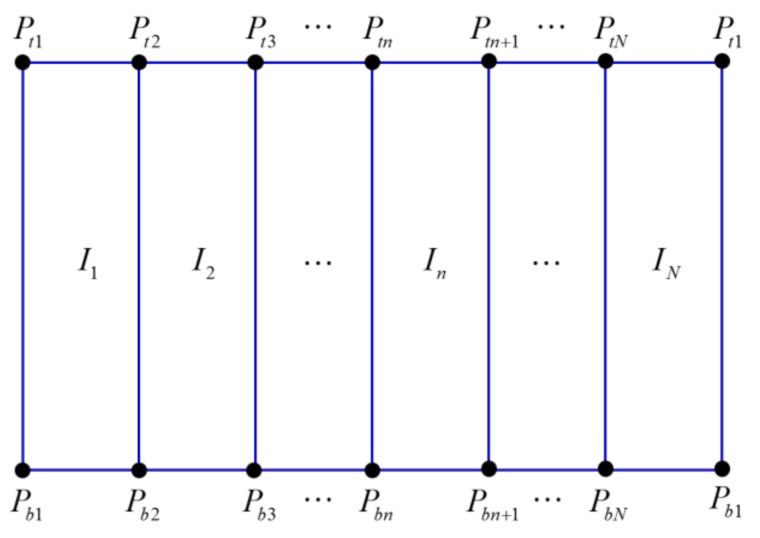
Splitting image *I* with three-dimensional points by Equations (7)–(10).

**Figure 10 sensors-19-00746-f010:**
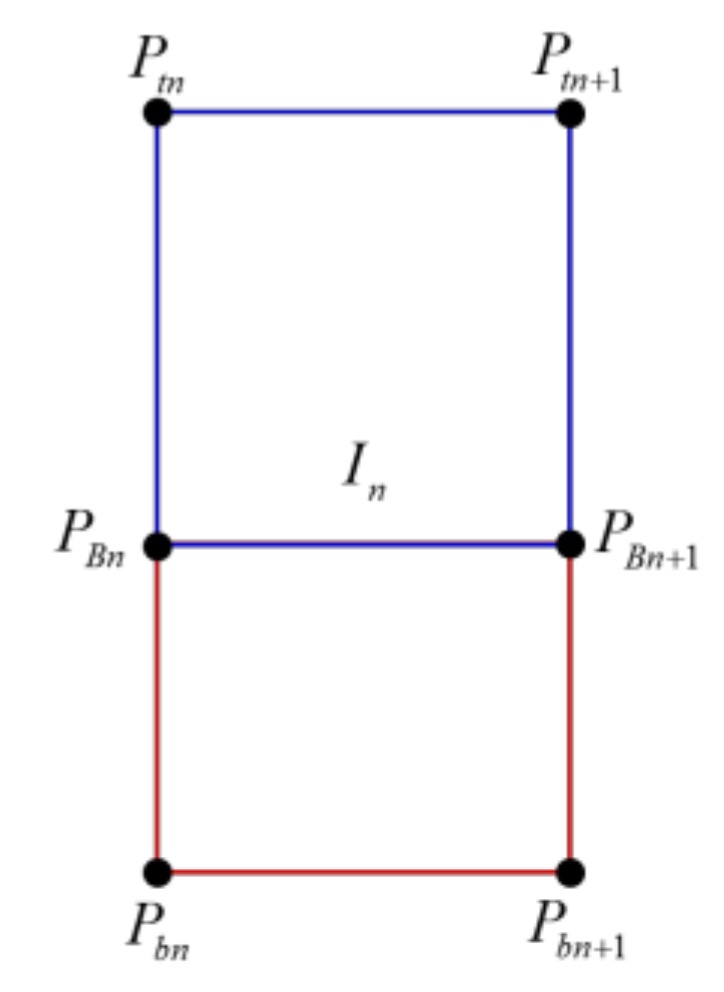
Splitting wall and floor images in *I**_n_*. The upper region (blue line) of the divided image is mapped to the wall, while the lower region (red line) is mapped to the blank floor.

**Figure 11 sensors-19-00746-f011:**
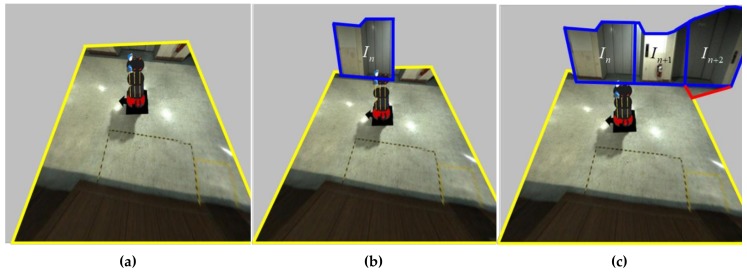
An example of mapping the stitched RoI A and B images to the spatial model. The regions of the yellow and blue lines are the RoI B and RoI A images in Figure 7. The region of the red line is the same as the region of the red line in Figure 10.

**Figure 12 sensors-19-00746-f012:**
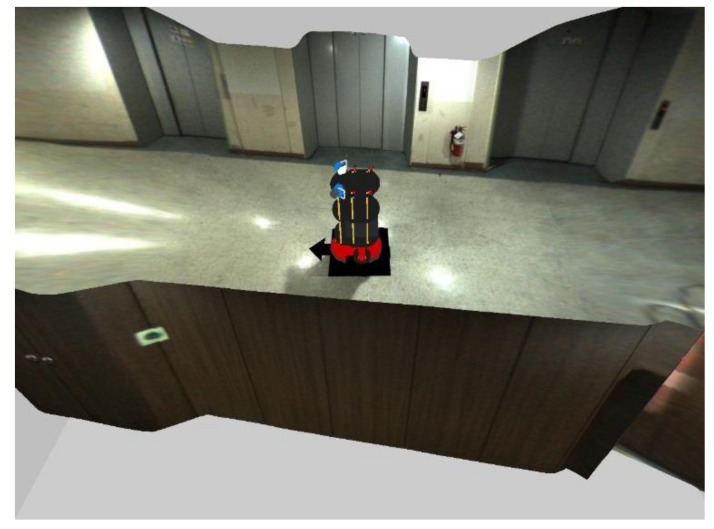
Results of mapping the stitched image to the spatial model in the corridor.

**Figure 13 sensors-19-00746-f013:**
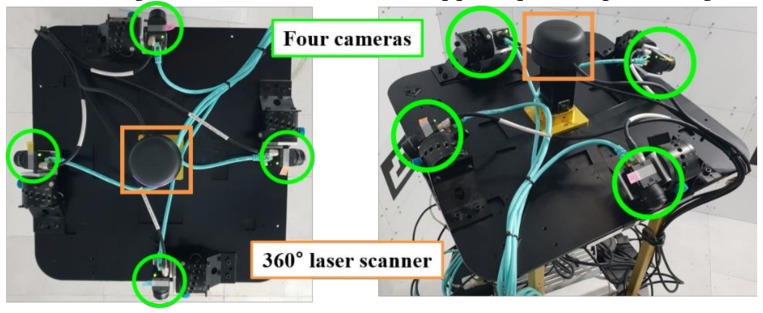
Configuration of four cameras and a 360° laser scanner on the robot.

**Figure 14 sensors-19-00746-f014:**
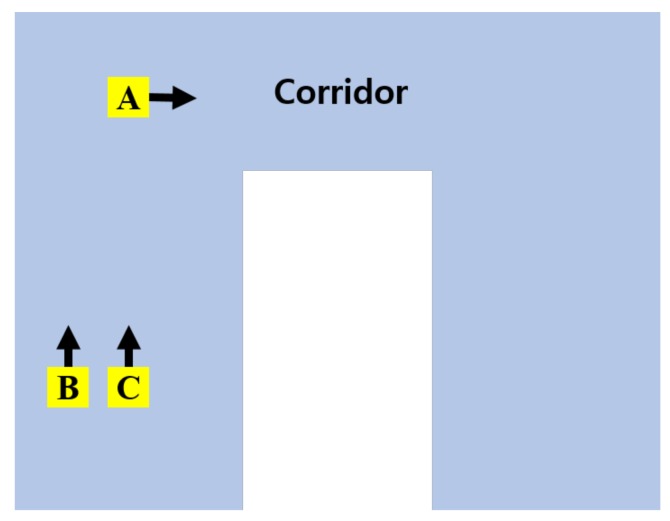
Robot’s location and direction (**A–C**) in the corridor drawing.

**Figure 15 sensors-19-00746-f015:**
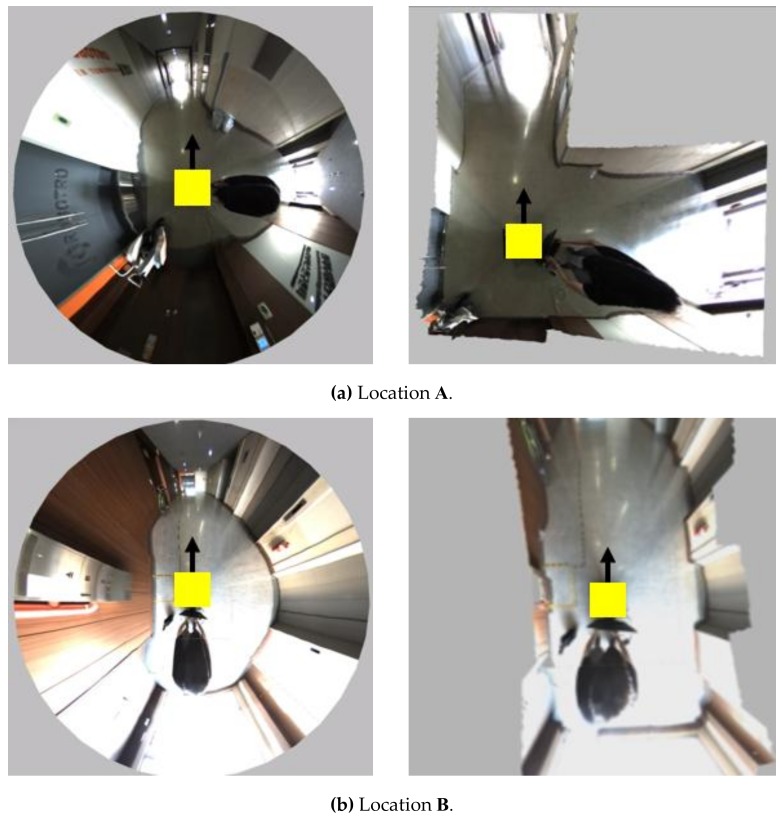
Comparison of the experimental results between the existing WAVM (left) [10] and the proposed system (right) at locations (**A–C**) in Figure 14.

**Figure 16 sensors-19-00746-f016:**
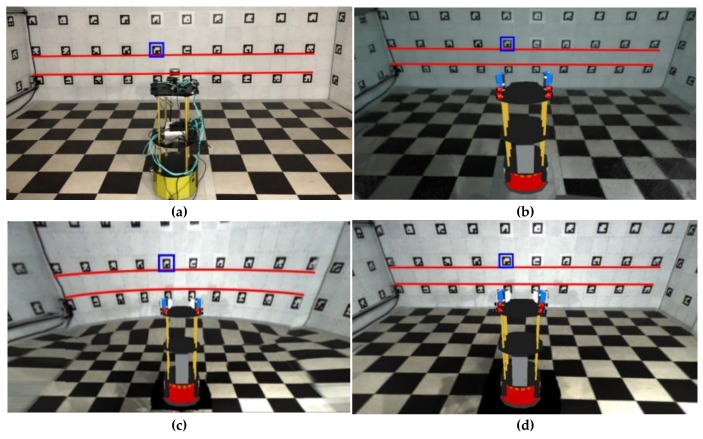
Result comparisons of four methods in the same viewpoint. (**a**) Ground truth, (**b**) SLAM [17,18,19], (**c**) existing WAVM [10], (**d**) proposed system.

**Figure 17 sensors-19-00746-f017:**
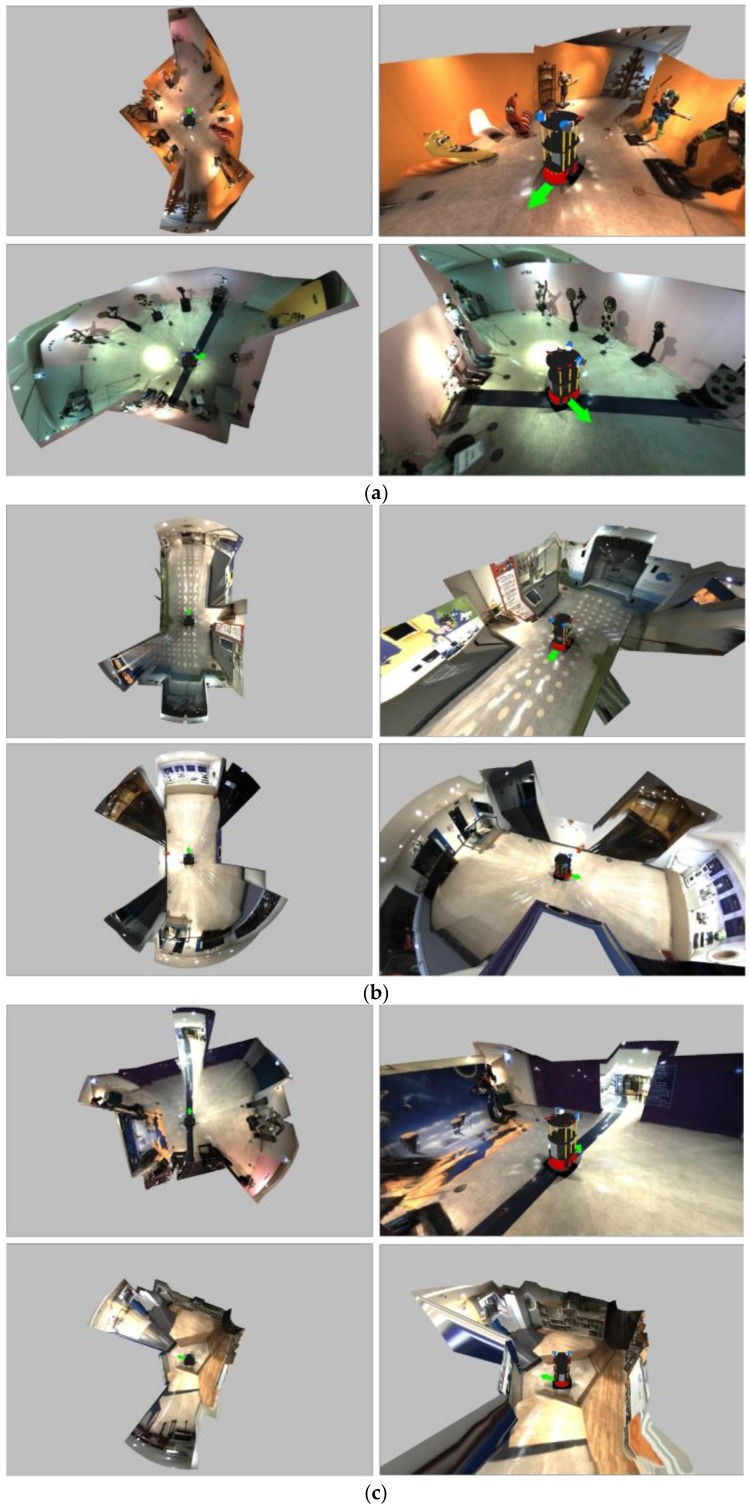
Resulting images in various and complex environments.

**Table 1 sensors-19-00746-t001:** Comparison of the root mean square error (RMSE) for the ground truth and results with methods. SLAM: simultaneous localization and mapping.

Methods	Distance		
	2 m	2.5 m	3 m
SLAM [17,18,19]	7.5894	8.1826	8.3263
Existing WAVM [10]	5.4782	5.6378	5.2512
Proposed system	3.9556	3.6842	3.8215

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
