# Peer review of "Three-Dimensional Visualization System with Spatial Information for Navigation of Tele-Operated Robots"

_sensors, 2019, doi:10.3390/s19030746_

Author Response

Point 1: The configuration of camera and laser scanner mounted on the robot must be clarified. So, a reviewer suggests that a figure showing the configuration should be included.

Response 1:

We added Figure 13 to the positions of the cameras and laser scanners mounted on the robot.                                            

Point 2: For clarifying the correspondence between the actual space and the results of the proposed method, the data of laser scanner and the image, (especially for corridors or rooms) of four cameras should be included in Section 3.1.

Response 2:

The resulting images in various and complex environments are shown in Figure 16.

Reviewer 2 Report

In this paper, the authors describe a three-dimensional visualization system using spatial information obtained by laser scanner for effectively controlling a tele-operated robot. There are several problems need to be further clarified in this manuscript.

It is shown that after undistortion operation, the images obtained by fisheye cameras are remain distorted, does the undistortion operation necessary? Is there a better scheme to correct the distorted images? 

Please further explain how to extract RoI regions from images. In the example shown in Figure 5, does the RoI A and B cropped manually? 

How to calculate the parameters in homography matrix? Are they keep change constantly?

How to determine the threshold D when mapping the stitched image to the spatial model? Whether more thresholds are needed under more complex situation?  

Since the stitched images are split into numerous images in mapping stage, for example, 720, how about the computational complexity? Please compare the time cost with other methods. 

It should be “split into N images” instead of “split into IN images” in the first paragraph of section 2.3.

Author Response

Point 1: It is shown that after undistortion operation, the images obtained by fisheye cameras are remain distorted, does the undistortion operation necessary? Is there a better scheme to correct the distorted images?

Response 1:

The reason why the image looks distorted after undistortion operation is that the image is not a flat front image but an image that includes floor, ceiling, and three walls (front, left and right sides).

In our system, (a) front view image and (b) bottom view image of Figure 6 are used and it can be confirmed that the distortion is removed.

Point 2: Please further explain how to extract RoI regions from images. In the example shown in Figure 5, does the RoI A and B cropped manually?

Response 2:

Different images are generated for each installed camera depending on the size of the sensor and the angle of view of the lens. Therefore, RoI A and B are set only once manually.

Point 3: How to calculate the parameters in homography matrix? Are they keep change constantly?

Response 3:

The homography matrix does not change constantly. As shown in Figure 5, RoI A uses corners between wall, wall, and ceiling, and RoI B uses floor check board to obtain homography parameters for a total of eight images.

Point 4: How to determine the threshold D when mapping the stitched image to the spatial model? Whether more thresholds are needed under more complex situation?

Response 4:

The threshold D has a value regardless of the complexity of the environment. D represents the distance value and is set to the actual distance to the upper side of the RoI B rectangle in Figure 5.

Point 5: Since the stitched images are split into numerous images in mapping stage, for example, 720, how about the computational complexity? Please compare the time cost with other methods.

Response 5:

N = 720 is the resolution of the 360°laser scanner to obtain spatial information, and the higher the resolution, the more accurate the spatial model. Also, the computational complexity increases in direct proportion (O(n)) as N increases.

Point 6: It should be “split into N images” instead of “split into IN images” in the first paragraph of section 2.3.

Response 6:

I have corrected everything you have pointed out.

Reviewer 3 Report

In this paper, it is difficult to find a clear novelty of the proposed solution. The introduction is missing other works in the subject area.  The article has a very poor review of similar solutions and is actually limited to two very basic proposals. The authors limited the described state of the problem to only two solutions (SLAM and WAVM methods). The presented solution consists a simple pipeline consisting of a three-stage process (image stitching, spatial modeling, and image mapping).  The entire pipeline of the proposed method consists of well-known and basic solutions.The article has a quite significant practical aspect but lacks a strong scientific side with a significant lack of scientific rigorousness, technical depth and analytical components. The assumptions used are generally acceptable. The superiority of the proposed method has been demonstrated but I am not satisfied with the results of the generality of the proposed method. The authors should conduct more experiments for different scenarios and different test scenes.

Author Response

Point 1: The authors should conduct more experiments for different scenarios and different test scenes

Response 1:

The resulting images in various and complex environments are shown in Figure 16.

Reviewer 4 Report

This is a well-written manuscript on an improved technical device for more natural visualization.

Author Response

Thank you for your comment.

Round  2

Reviewer 2 Report

I suggest this revised manuscript can be published.

Reviewer 3 Report

answer and changes made by the authors in relation to the previous form are enough for me.